# Sex differences in discrimination behavior and orbitofrontal engagement during context-gated reward prediction

**Sophie Peterson[1†], Amanda Maheras[2†], Brenda Wu[1], Jose Chavira[1], Ronald Keiflin[1,3]***

[1]Department of Psychological & Brain Sciences, University of California, Santa Barbara, Santa Barbara, United States; [2]Department of Molecular, Cellular & Developmental Biology, University of California, Santa Barbara, Santa Barbara, United States; [3]Neuroscience Research Institute, University of California, Santa Barbara, Santa Barbara, United States

**\*For correspondence:**
rkeiflin@ucsb.edu

[†]These authors contributed equally to this work

**Competing interest:** The authors declare that no competing interests exist.

**Abstract** Animals, including humans, rely on contextual information to interpret ambiguous stimuli. Impaired context processing is a hallmark of several neuropsychiatric disorders, including schizophrenia, autism spectrum disorders, post-traumatic stress disorder, and addiction. While sex differences in the prevalence and manifestations of these disorders are well established, potential sex differences in context processing remain uncertain. Here, we examined sex differences in the contextual control over cue-evoked reward seeking and its neural correlates, in rats. Male and female rats were trained in a bidirectional occasion-setting preparation in which the validity of two auditory reward-predictive cues was informed by the presence, or absence, of a visual contextual feature (LIGHT: X+/DARK: X−/LIGHT: Y−/DARK: Y+). Females were significantly slower to acquire contextual control over cue-evoked reward seeking. However, once established, the contextual control over behavior was more robust in female rats; it showed less within-session variability (less influence of prior reward) and greater resistance to acute stress. This superior contextual control achieved by females was accompanied by an increased activation of the orbitofrontal cortex (OFC) compared to males. Critically, these behavioral and neural sex differences were specific to the contextual modulation process and not observed in simple, context-independent, reward prediction tasks. These results indicate a sex-biased trade-off between the speed of acquisition and the robustness of performance in the contextual modulation of cued reward seeking. The different distribution of sexes along the fast learning ↔ steady performance continuum might reflect different levels of engagement of the OFC, and might have implications for our understanding of sex differences in psychiatric disorders.

## eLife assessment

This **valuable** manuscript reveals sex differences in bi-conditional Pavlovian learning and conditional behavior. Males learn hierarchical context-cue-outcome associations more quickly, but females show more stable and robust task performance. These sex differences are related to cellular activation in the orbitofrontal cortex. Although the evidence supporting these claims is **convincing**, some assertions of sex differences in context-dependent discrimination behaviour may be slightly overstated yet have strong potential to guide future research to clarify the nature of these differences. The results will be of interest to many behavioural neuroscientists, particularly those who investigate sex-specific behaviours.

## Introduction

Pavlovian learning is often conceived as the formation of a binary association between two events, where event #1 (the stimulus) evokes the anticipation of event #2 (the outcome). Under this naive definition, the background circumstances (the context) in which those events occur play little role. However, this simple form of learning is often augmented by higher-level cognitive systems that allow animals (including humans) to learn and retrieve context-specific associations (e.g., the word '*shot*' might evoke different representations in a bar vs. a vaccine clinic) (*Gershman, 2017*; *Swartzentruber, 1995*; *Fraser and Holland, 2019*). This process is referred to as 'contextual-gating' (the context acting as a 'gatekeeper' in the selection of the appropriate associative memory).

Deficits in contextual gating are observed in several neuropsychiatric disorders including schizophrenia, autism spectrum disorder (ASD), post-traumatic stress disorders (PTSD), and substance use disorders (SUD) (*Maren et al., 2013*; *Vermeulen, 2015*; *Elman et al., 2023*; *Hemsley, 2005*; *Servan-Schreiber et al., 1996*; *Liberzon and Abelson, 2016*; *Frith and Happé, 1994*; *Jones et al., 2013*; *Chatham et al., 2012*), and may contribute to the intrusive thoughts and maladaptive (situation-inappropriate) behaviors that characterize these disorders. Importantly, the prevalence and symptomatology of these disorders vary between sexes. For example, the incidence of schizophrenia and ASD is higher in men (*Santos et al., 2022*) while women are more at risk of developing SUD or PTSD following initial drug use or trauma (*Towers et al., 2023*; *Tolin and Foa, 2006*). This disparity might be due, in part, to subtle sex differences in executive strategies (*Baron-Cohen, 2002*; *Grissom and Reyes, 2019*; *Geary, 2019*; *Andreano and Cahill, 2009*). Stress, a contributing factor in all aforementioned disorders, can further exacerbate these sex differences, resulting in sex-specific cognitive vulnerabilities (*Geary, 2019*; *Andreano and Cahill, 2009*; *Goldfarb et al., 2019*; *Mazure et al., 2023*; *Bangasser et al., 2018*; *Klein and Corwin, 2002*). Sex differences in executive functions and sensitivity to stress have been reproduced in animal models using behavioral assays of attention, impulsivity, spatial navigation, and working memory (*Grissom and Reyes, 2019*; *Orsini and Setlow, 2017*; *Shansky et al., 2004*; *Papaleo et al., 2012*). Whether these sex differences extend to the contextual gating of associative predictions remains unknown.

In animal models, the contextual gating of associative predictions can be studied in occasion-setting preparations. In positive occasion-setting, animals learn that a 'target' cue (X) results in a reward outcome (+) only when that cue is accompanied by a specific 'feature' such as a specific context (A); the same cue presented in absence of this contextual feature remains without consequence (A:X+/X−). Conversely, in negative occasion-setting, a target cue results in reward, except when accompanied by a specific feature (A:X−/X+). In both preparations, the contextual feature is thought to hierarchically modulate (positively or negatively) the association between the target cue and the outcome (*Fraser and Holland, 2019*; *Bonardi et al., 2017*). Strong evidence for this hierarchical modulation is the fact that a contextual feature can simultaneously function as a positive modulator for one target cue and as a negative modulator for another target cue (A:X+/X−:A:Y−/Y+). In such bidirectional occasion-setting procedures all cues and contexts have equal probabilities of reward, therefore, accurate reward predictions cannot be achieved via simple associative summation but instead require the context-informed hierarchical gating of associative predictions (i.e., context-gated associative predictions).

Compared to simple forms of associative learning, the neural bases of context-gated associative predictions (or occasion-setting) remain severely understudied. Although recent studies have implicated the orbitofrontal cortex (OFC) in occasion-setting (*Fraser and Janak, 2023*; *Meyer and Bucci, 2016*; *Shobe et al., 2017*), most of the behavioral and neurobiological occasion-setting studies have exclusively used male subjects and/or did not explicitly address sex as a biological variable.

Therefore, the purpose of this study was twofold: (1) investigate sex differences in the (bidirectional) contextual gating of reward prediction and its sensitivity to stress, and (2) investigate sex differences in OFC engagement during the contextual gating of reward predictions. We found that female rats were slower than males to acquire context-gated reward seeking. However, once established, context-gated performance was more stable in females; it showed less variability within a session and was more resistant to acute stress. Moreover, we showed that this superior contextual control achieved by females was accompanied by an increased activation of the OFC.

# Results

Male and female rats were randomly assigned to one of four Pavlovian discrimination tasks outlined in *Figure 1A–C*. In these tasks [adapted from *Delamater et al., 2010*], all rats were exposed to two auditory cues (X and Y, 10 s) presented within two distinct, alternating, visual contexts (LIGHT and DARK; 2 min). All rats received similar amount of reward, and produced a similar behavioral response (magazine approach). However, the different reward contingencies featured in these tasks presumably promoted the emergence of different associative architectures for reward prediction (*Figure 1B*). In the **Ctx-dep. O1** task, cue X was rewarded only in the LIGHT context and cue Y was rewarded only in DARK context; all rewarded trials resulted in the same outcome (O1). Accurate reward prediction in this task critically relies on the context-informed hierarchical modulation associative predictions (i.e., occasion-setting). In the **Simple discrimination** task, cue X was always rewarded and cue Y was never rewarded, regardless of context. In the **No discrimination** task both cues X and Y were probabilistically rewarded in both contexts. Finally, the **Ctx-dep. O1/O2** task featured similar contingencies as *Ctx-dep. O1*, with the exception that the two rewarded trials (LIGHT: X and DARK: Y) resulted in two different outcomes (O1 and O2, respectively). In this case, not only might the visual context promote the hierarchical modulation of associative predictions (as in *Ctx-dep. O1*), but the context might also contribute to context-sensitive performance via simple (nonhierarchical) associations. Indeed, the use of differential outcomes allows for distinct context–outcome associations (LIGHT → O1 and DARK → O2), as well as distinct cue–outcome associations (X → O1 and Y → O2). On a given trial, if the cue and context tend to activate the same outcome representation, the converging cue- and context-evoked excitations can add up. As a result, context-sensitive performance could potentially be achieved via simple (nonhierarchical) summation process. Therefore, while rats trained in *Ctx-dep. O1/O2* might engage a combination of associative processes to achieve context-sensitive behavior (including hierarchical and nonhierarchical associations), only rats trained in the *Ctx-dep. O1* task critically and unambiguously rely on hierarchical associations to achieve context-sensitive behavior.

## Male rats acquire context regulation over reward seeking faster than females

*Figure 1D* displays the course of acquisition of discriminated Pavlovian approach in all four tasks (*Ctx-dep. O1 n* = 30F + 32M; *Simple discrimination n* = 12F + 22M; *No discrimination n* = 8F + 8M; *Ctx-dep. O1/O2 n* = 14F + 14M). In the context-dependent discrimination with a single outcome (*Ctx-dep. O1*), Linear Mixed Models (LMM) analysis revealed a significant Trial × Session interaction ($F(17, 181.488) = 73.343$, $p < 0.001$), as rats gradually learned to respond more during rewarded than non-rewarded trials. Critically, the analysis also revealed a significant Sex × Trial × Session interaction ($F(17, 181.488) = 4.683$, $p < 0.001$). Post hoc comparisons revealed that males and females displayed similar levels of responding to rewarded trials (ps > 0.260), but females maintained higher responding during non-rewarded trials, causing both sexes to differ in this measure from session 40 onwards (ps < 0.047). To facilitate comparisons between groups, we calculated for each animal a discrimination ratio, defined as the time in port during rewarded trials divided by the total time in port during all trials (*Figure 1E*). With the exception of animals trained in the *No Discrimination* task (for which discrimination between rewarded and non-rewarded trials was impossible by design), all training groups demonstrated a gradual increase in discrimination ratio (main Session effect: *No Discr.* p = 0.477; *all other groups* ps < 0.001). Importantly, only in *Ctx-dep. O1* did we detect a main effect of Sex ($F(1, 69.876) = 15.185$, $p < 0.001$) and a Sex × Session interaction ($F(17, 365.569) = 3.054$, $p < 0.001$). Indeed, acquisition of discriminated performance was faster in males, causing significant sex differences from session 28 onwards (ps < 0.014). No main effect or interaction with Sex was detected for any other training group (ps > 0.305).

 *Figure 2A* summarizes the performance of each group (time in port during the four trial types, LIGHT: X/DARK: X/LIGHT: Y/DARK: Y) by the end of the acquisition period (i.e., after 72 sessions for *Ctx-dep. O1*, or 32 session for all other groups). We next calculated the proportion of male and female rats that successfully acquired discriminated responding during the training period. The criterion for successful discrimination was defined as a time in port during rewarded trials that exceeded the time in port during non-rewarded trials by at least 50% (Time $_{rew}$ ≥1.5* Time $_{nonrew}$, which corresponds to a discrimination ratio ≥0.6) for a minimum of four out of five consecutive sessions. Rats in the Ctx-dep. discrimination groups had to reach this criterion for both cues to be considered 'discriminators'

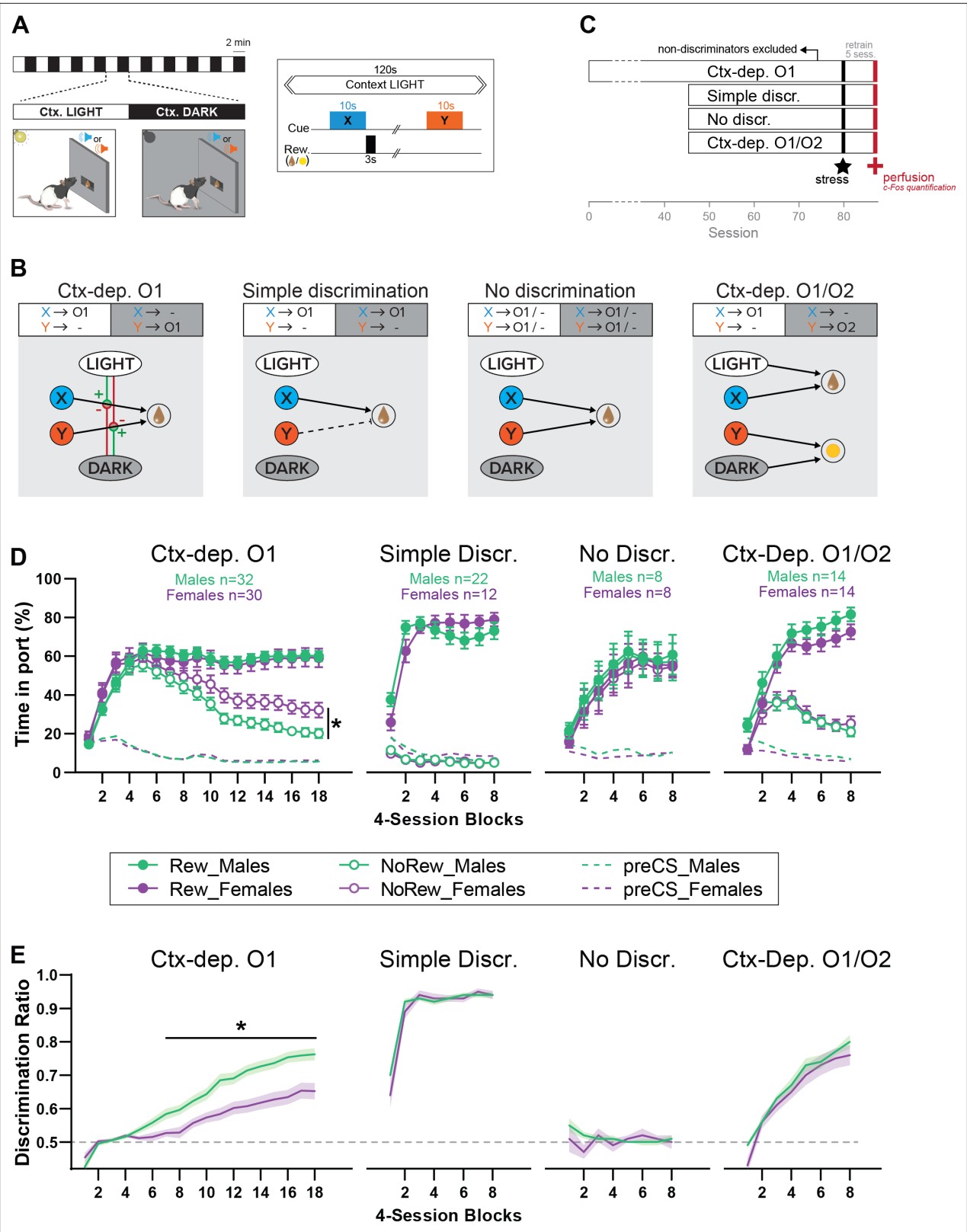

**Figure 1.** Sex differences in the acquisition of context-gated but not context-independent reward predictions. (**A**) Apparatus and general task design. During each session, the local context alternates every 2 min between a Ctx. LIGHT (houselight on) and Ctx. DARK (houselight off). Embedded in these context cycles, brief auditory stimuli (X or Y; 10 s) are presented and potentially followed by reward (chocolate milk or banana pellets). (**B**) Reward contingencies and corresponding associative structures for each training condition. *Ctx-dep. O1*: the local context informs (hierarchically 'gates') the

*Figure 1 continued on next page*

*Figure 1 continued*

predictive status of the ambiguous reward cues X and Y. *Simple discrimination*: reward cues are not ambiguous (only X is rewarded) and the local context is irrelevant. *No Discrimination*: reward cues X and Y are potentially rewarded and the local context is irrelevant. *Ctx-dep. O1/O2*: context-dependent discrimination with differential outcomes; in this case, the local context is relevant but may contribute to context-sensitive performance via simple (nonhierarchical) associations (summation of cue- and context-evoked predictions). (**C**) Experimental timeline. (**D**) Time in port (% cue time) during rewarded and non-rewarded trials across acquisition for male and female rats trained in each task. (**E**) Discrimination ratio across acquisition for male and female rats trained in each task. Error bars and error bands indicate ± standard error of the mean (s.e.m.). *p < 0.05, *t*-tests.

The online version of this article includes the following source data for figure 1:

**Source data 1.**

(LIGHT:X ≥1.5* DARK:X and DARK:Y ≥1.5* LIGHT:Y). A significant sex difference in the proportion of discriminators was observed in the *Ctx-dep. O1* group ($\chi^2$ = 5.35; p = 0.021), as a greater proportion of males reached criterion (***Figure 2B, C***). No sex differences were observed in the proportion of discriminators in any other tasks.

We then analyzed the number of sessions required for rats to reach the discrimination criterion (***Figure 2D***). Rats that failed to reach the discrimination criterion were excluded from this analysis. A Kruskal–Wallis one-way analysis of variance (ANOVA) revealed a significant effect of the behavioral task (*H*(2) = 87.361, p < 0.001) with different sessions to criterion for all the discrimination tasks (*Ctx-dep. O1 > Ctx dep. O1/O2 > Simple*; ps < 0.001, Dunn's tests). Note how the introduction of differential outcomes profoundly facilitated the acquisition of context-dependent discrimination (sessions to criterion: 47.8 ± 1.7 standard error of the mean [s.e.m.] for *Ctx-dep. O1* vs. 17.26 ± 1.6 s.e.m. for Ctx dep. O1/O2). This is consistent with a prior study showing a similar differential outcome effect in this task [***Delamater et al., 2010***].

Sex differences in the speed of acquisition (i.e., sessions to criterion) were analyzed separately within each task. In *Ctx-dep. O1*, males were significantly faster to reach the discrimination criterion (*T*(38) = 2.502, p = 0.017). No sex differences in the speed of acquisition were observed in the other tasks (ps ≥ 0.175).

These results indicate that female rats were generally slower to acquire contextual control over cue-evoked reward seeking. This sex difference in the development of discriminated performance was observed only in *Ctx-dep. O1*, the only task that unambiguously engages contextual modulation of reward predictions. Note that estrous cycle did not affect performance in female discriminators (***Figure 2—figure supplement 1***). Rats that failed to reach the discrimination criterion at the end of the acquisition period were excluded from further experimental procedures (with the exception of rats trained in the *No Discrimination* group).

## Trial history influences context-gated reward seeking in male but not female rats

The delivery of reward, while critical for learning, can also interfere with the expression of discrimination behavior (***Kuchibhotla et al., 2019***). Therefore, we investigated to which extent within-session reward history (i.e., prior rewarded or non-rewarded trials) influenced performance in the *Ctx-dep. O1* task, in male (*n* = 25) and female (*n* = 15) discriminators (***Figure 3***). Performance was analyzed over 10 consecutive sessions (sessions 71–80; 80 trials/session; 800 trials total), a period during which discrimination performance was globally stable. The first trial of a session was excluded as it was not preceded by any trial. For the remaining 790 trials (10 sessions × 79 trials), we determined if the trial was preceded by a rewarded or non-rewarded trial. This large sample size is required to ensure that enough data were collected for each possible trial history scenario. Responding during non-rewarded and rewarded trials was analyzed separately.

***Figure 3A*** shows the time in port during non-rewarded trials, as a function of the outcome of the previous trial (prior reward, or non-reward). A repeated measures (RM)-ANOVA revealed a main effect of Prior Rewards (*F*(1, 38) = 27.144; p < 0.001) as prior rewards increased responding during subsequent non-rewarded trials. The analysis showed no main effect of Sex (*F*(1, 38) = 1.783; p = 0.190), but a marginal Sex × Prior Reward interaction (F(1, 38) = 4.015; p = 0.052), as males were slightly more influenced by prior rewards. The effect of trial history appears to be carried exclusively by the outcome of the previous trial (*t* − 1) (no effect of consecutive prior rewards or non-rewards; ps ≥ 0.147, Bonferroni *t*-tests; ***Figure 3B***).

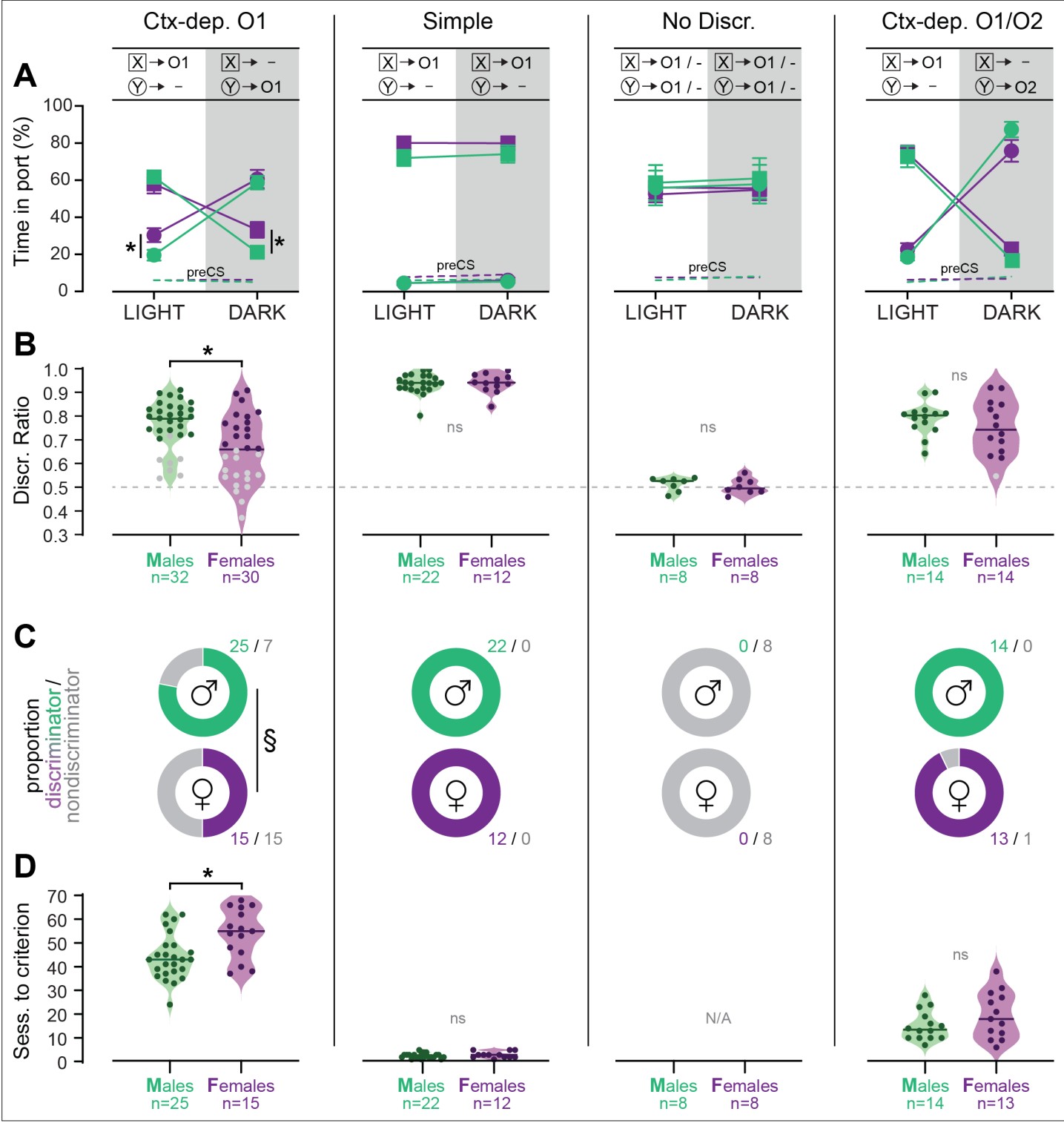

**Figure 2.** Individual differences in the acquisition of discriminated Pavlovian approach. (**A**) Time in port (% cue time) during the four different trial types (LIGHT: X/DARK: X/LIGHT: Y/DARK: Y) at the end of the training period (last 3 days average) for male and female rats trained in each discrimination task. (**B**) Discrimination ratio at the end of the training period (last 3 days average) for male and female rats trained in each discrimination task. Dark symbol: discriminator; gray symbol: non-discriminator. Horizontal bar = group median; dashed horizontal line represents the indifference level (no discrimination). (**C**) Proportion of male and female rats classified as discriminator or non-discriminator in each task. (**D**) Speed of acquisition (sessions to criterion) for discriminator rats, in each task. Horizontal bar = group median. *p < 0.05 males vs. females, *t*-tests; §p < 0.05 chi-square test.

The online version of this article includes the following source data and figure supplement(s) for figure 2:

*Figure 2 continued on next page*

*Figure 2 continued*

**Source data 1.**

**Figure supplement 1.** Estrus cycle does not influence the expression of context-gated reward seeking.

**Figure supplement 1—source data 1.**

We then analyzed another metric of conditioned responding: the response probability (i.e., probability subject enters the reward port during cue presentation). We reasoned that this metric might be more prone to detect brief lapses in discrimination performance (false alarms). *Figure 3C* shows the probability of responding to a non-rewarded cue as a function of the outcome of the prior trial (prior reward or non-reward). A RM-ANOVA revealed a main effect of Prior Reward ($F(1, 38) = 25.607$; $p < 0.001$), and a significant Sex × Prior Reward interaction ($F(1, 38) = 9.247$; $p = 0.004$). In males, prior reward increased the probability of responding on subsequent non-rewarded trials ($T(38) = 43.752$; $p < 0.001$) while in females, trial history had no effect ($T(38) = 1.631$; $p = 0.209$). As a result, males and females differed in their probability of responding to non-rewarded trials when those trials were preceded by reward ($T(38) = 4.508$; $p = 0.040$). Here again we found no influence of the number of consecutive past reward or non-rewards (ps ≥0.710, *Figure 3D*).

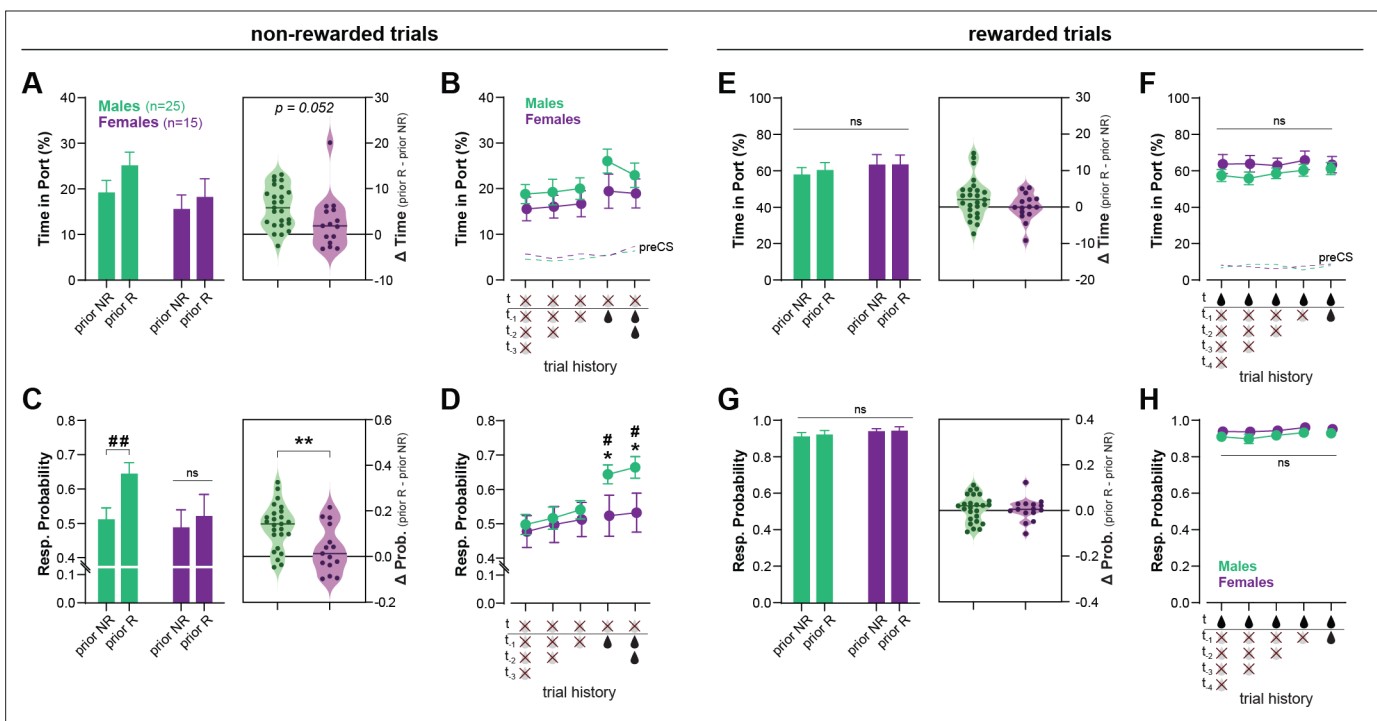

**Figure 3.** Recent rewards disrupt contextual control of reward seeking in male, but not female, rats. (**A**) Time in port (% cue time) during non-rewarded trials as a function of prior trial (rewarded: prior R, or non-rewarded: prior NR) for male and female rats trained in *Ctx-dep. O1* task. Inset: increase in responding caused by prior reward for male and female rats (horizontal bar = median difference score). (**B**) Time in port during non-rewarded trials as a function of prior consecutive non-rewarded or rewarded trials. (**C**) Response probability on non-rewarded trials as a function of prior trial (rewarded: prior R, or non-rewarded: prior NR). Inset: increase in the probability of responding caused by prior reward for male and female rats (horizontal bar = median difference score). (**D**) Response probability on non-rewarded trials as a function of prior consecutive non-rewarded or rewarded trials. (**E**) Time in port (% cue time) during rewarded trials as a function of prior trial (rewarded: prior R, or non-rewarded: prior NR). Inset: difference in responding caused by prior reward for male and female rats. (**F**) Time in port during rewarded trials as a function of prior consecutive non-rewarded or rewarded trials. (**G**) Response probability on rewarded trials as a function of prior trial (rewarded: prior R, or non-rewarded: prior NR). Inset: difference in the probability of responding caused by prior reward for male and female rats. (**H**) Response probability on rewarded trials as a function of prior consecutive non-rewarded or rewarded trials. Error bars indicate ± standard error of the mean (s.e.m.). *p < 0.05; **p < 0.01 males vs. females; #p < 0.05; ##<0.01 prior R vs. prior N; *t*-tests.

The online version of this article includes the following source data for figure 3:

**Source data 1.**

Responding to rewarded cues remained high throughout the sessions and was not influenced by trial history (ps ≥ 0.113); moreover no main effect or interaction with Sex was observed on these rewarded trials (ps ≥ 0.127) (*Figure 3E–H*).

These results indicate that male rats have difficulty inhibiting responding to non-rewarded (contextually irrelevant) cues following a recent reward. In contrast, female rats maintained high discrimination performance throughout the session and recent rewards had little to no effect on their discrimination accuracy. The mechanisms for the reward-induced decrease in discrimination accuracy (here observed primarily in males) are not fully understood. Rewards might temporarily lower the threshold for behavioral responding, resulting in stronger responding to weakly predictive cues (*Kuchibhotla et al., 2019*). Additionally, reward delivery might decrease discrimination accuracy by increasing the predictive value of the cue (and context) present during reward delivery. These direct cue → outcome (and context → outcome) associations compete with hierarchical associations for the control of behavior (*Bradfield and Balleine, 2013*), potentially affecting responding on the next trial (e.g., a rewarded LIGHT: X+ trial might increase the value of cue X and promote responding during the following DARK: X− trial; the same logic could be applied to the influence of the context). Further investigation is required to determine the precise mechanism(s) by which reward disrupt discrimination performance in the context-dependent discrimination task.

## Acute stress disrupts context-gated reward seeking in male but not female rats

Stress affects several cognitive processes (e.g., selective attention, spatial navigation, working memory), often in a sex-dependent manner (*Papaleo et al., 2012*; *Beiko et al., 2004*; *Shansky et al., 2006*). However, the effect of stress on contextual modulation, and possible sex-differences in this effect, have not been formally tested. Therefore, we tested the effect of an acute stress (90 min restrain) on discrimination performance (expressed as discrimination ratio), in a subset of animals from all training groups (*Ctx-dep O1*: n = 11F + 15M; *Simple discrimination*: n = 6F + 6M; *No discrimination*: n = 8F + 8M; *Ctx-dep O1/O2*: n = 7F + 7M) (*Figure 4*). With the exception of the *No Discrimination* group, acute stress reduced discrimination accuracy in all tasks (main Stress effect: ps ≤ 0.004). However, only in the *Ctx-dep. O1* task did we observe a significant Stress × Sex interaction ($F_{(1, 24)}$ = 5.665; p = 0.026) as only males showed reduced discrimination performance following acute stress (Stress vs. Baseline; males: $T_{(14)}$ = 5.165; p < 0.001; females: $T_{(10)}$ = 1.290; p = 0.209). No main effect or interaction with Sex was observed in any other task (ps ≥ 0.178; note however the smaller sample size in all these other groups). Therefore, although acute stress disrupted discrimination performance in all tasks, females trained in *Ctx-dep. O1* appear to be less sensitive to the disrupting effect of stress in this task. This is despite the fact that female rodents typically show greater hypothalamo–pituitary–adrenal (HPA) activation in response to this type of stressor (i.e., greater stress activation) (*Babb et al., 2013*; *Lovelock and Deak, 2020*).

## Female rats show higher orbitofrontal engagement during context-gated reward predictions

The hierarchical gating of associative predictions has been linked to activity in the OFC (*Fraser and Janak, 2023*; *Meyer and Bucci, 2016*; *Shobe et al., 2017*). Therefore, for a subset of animals (*Ctx-dep. O1*: n = 7F + 9M; *Simple discrimination*: n = 7F + 9M; *No discrimination*: n = 7F + 7M; *Ctx-dep. O1/O2*: n = 5F + 5M), we quantified c-Fos-immunoreactive (cfos+) cells in the OFC (medial, ventral, and lateral subregions) following regular behavior testing (no-stress condition). We also quantified cfos+ in two control regions, M1J and S1J, where we did not expect task-specific activation. Compared to baseline (homecage controls), behavioral testing increased cfos+ expression in all regions of interest, regardless of task (*Figure 5*; homecage c-Fos+ counts are shown but not included in the following statistical analyses). A RM-ANOVA revealed a main effect of Region ($F_{(4, 192)}$ = 85.037; p ≤ 0.001) and a significant Task × Sex × Region interaction ($F_{(12, 192)}$ = 1.9; p = 0.036). Follow-up planned contrast analysis compared males and females within each task and brain region. In the *Ctx-dep. O1* task, females showed increased c-fos+ expression compared to males, in all OFC subregions ($T_{(14)}$ ≥ 2.387; p ≤ 0.032). No sex differences were found in any other task (ps ≥ 0.255; note however the smaller sample size of the *Ctx-dep. O1/O2* group). Moreover, no sex differences were found in the control regions M1J and S1J, regardless of task (ps ≥ 0.139).

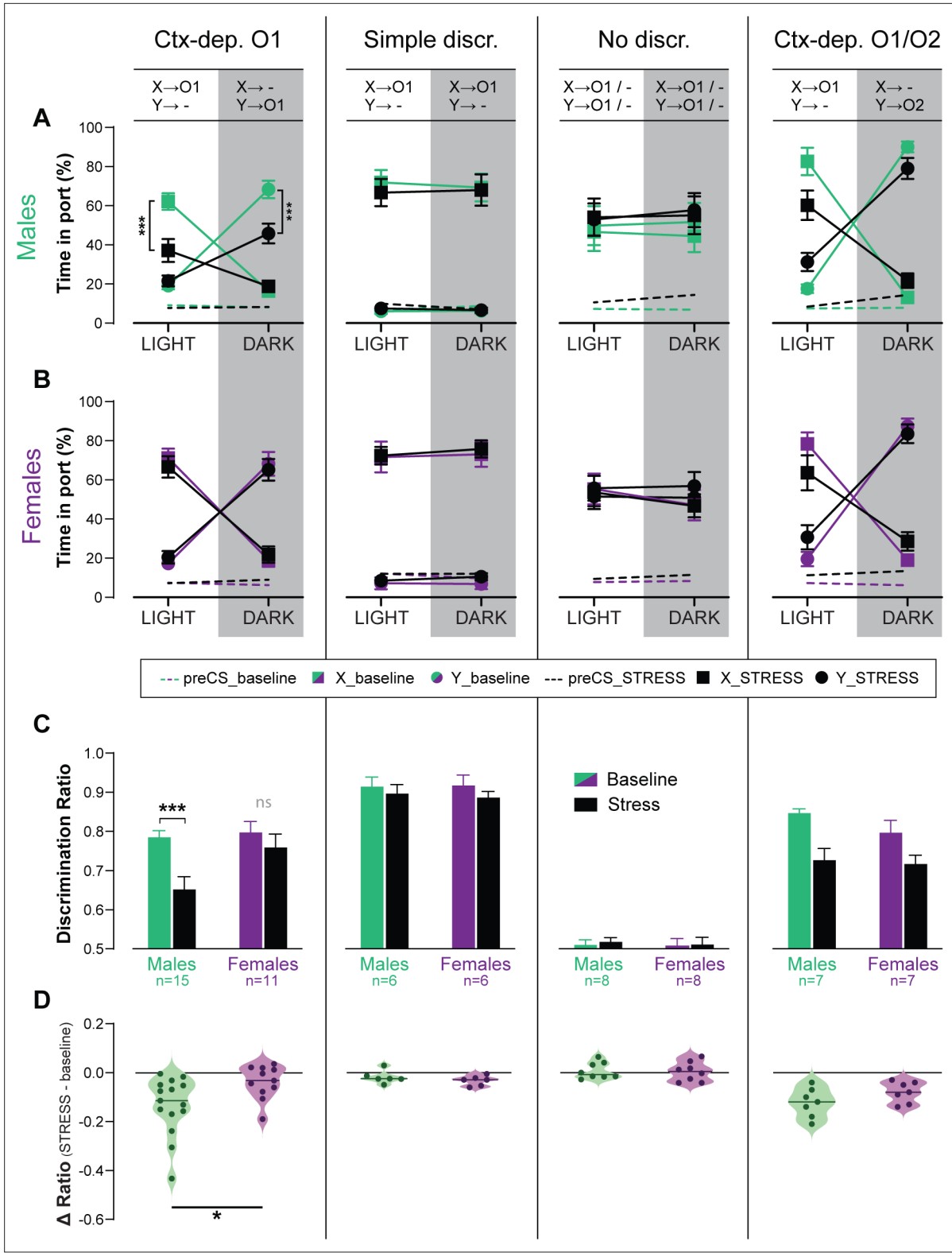

**Figure 4.** Acute stress disrupts context-gated reward seeking in male but not female rats. Time in port (% cue time) during the four different trial types (LIGHT: X/DARK: X/LIGHT: Y/DARK: Y) at baseline and after acute stress for male (**A**) and female (**B**) rats trained in each discrimination task. (**C**) Discrimination ratio at baseline and after acute stress, for male and female rats trained in each discrimination task. (**D**) Stress-induced disruption in discrimination performance (stress – baseline) for male and female rats trained in each discrimination task (horizontal bar = median difference score). Error bars indicate ± standard error of the mean (s.e.m.). *p < 0.05; ***p < 0.001, *t*-tests.

*Figure 4 continued on next page*

*Figure 4 continued*

The online version of this article includes the following source data for figure 4:

**Source data 1.**

## Discussion

We investigated sex differences in contextual modulation of cue-evoked reward predictions and its neural correlates. Male and female rats were trained on a context-dependent discrimination task (*Ctx-dep. O1*) in which a contextual feature informs the validity of two reward cues (i.e., bidirectional occasion-setting). Overall, males acquired context-gated behavior faster than females. However, once established, contextual control over behavior was more robust in females; it was more stable within a session (less influence of prior reward) and more resistant to acute stress. This superior contextual control achieved by females was accompanied by elevated c-Fos expression in all OFC subregions (medial, ventral, and lateral). Critically, no sex differences (behavioral or neurobiological) were observed among animals trained on simpler discrimination tasks that do not engage contextual gating (*Simple Discrimination*, *No discrimination*, *Ctx-dep. O1/O2*). These results add to the growing body of literature on sex differences in motivated behaviors and value-based decision-making (*Grissom and Reyes, 2019*; *Orsini and Setlow, 2017*). Most of these sex differences were observed in behaviors informed by binary associations between cues (or actions) and rewards. To our knowledge, this study is the first to demonstrate sex differences in the context-informed hierarchical gating of associative predictions, a process that is largely independent from binary associations (*Fraser and Holland, 2019*; *Bonardi et al., 2017*).

For *Ctx-dep. O1*, the context (LIGHT or DARK chamber) is an occasion-setter, that is, a stimulus that modulates the response to a target cue, presumably by gating the retrieval of the appropriate cue → outcome association. However, our task differs from classic occasion-setting preparations in several ways. Historically, Pavlovian modulation has been studied in preparations in which the modulating stimuli (the occasion-setter) exerts unidirectional modulation, either potentiating (A:X+/X−) or reducing (A:X−/X+) the response to the target cue. In such preparations, the occasion-setter might also establish a direct association (excitatory or inhibitory) with the outcome (*Gershman, 2017*; *Urcelay and Miller, 2014*; *Brandon and Wagner, 1991*), thereby entangling the contributions of hierarchical modulation and direct stimuli-evoked predictions. To circumvent this issue, occasion-setting preparations commonly introduce a temporal gap (5–30 s) between the termination of a discrete occasion-setting stimulus and its target cue. This approach has generally been successful in limiting direct associations between the occasion-setter and the outcome, instead promoting hierarchical modulation processes (*Ross and Holland, 1981*; *Holland, 1998*; but see *Bowers and Timberlake, 2017*). This temporal gap however introduces a working-memory requirement (animals must remember if the target cue was preceded by the occasion-setter) which complicates results interpretations.

Here, we used a bidirectional occasion-setting preparation with a single reward outcome (*Ctx-dep. O1*) as an alternative approach to investigate contextual modulation (*Delamater et al., 2010*). In this paradigm, a contextual feature (e.g., LIGHT) positively modulates one target association (LIGHT: X+/DARK: X−) and negatively modulates another association (LIGHT: Y−/DARK: Y+). Critically, there are no direct informative associations between contextual feature and the single reward outcome (rewards are equally frequent in the LIGHT or DARK contexts), so accurate discrimination necessitates hierarchical modulation (*Bonardi et al., 2017*). Moreover, the contextual feature precedes and overlaps with the target cue, thereby eliminating working-memory requirements. Thus, sex differences observed in *Ctx-dep. O1*, strongly indicate sex differences in hierarchical (context-informed) associative modulations. Alternatively, males and females might differ in terms of attentional resources (*Jentsch and Taylor, 2003*). Indeed, unlike *Simple Discrimination* or *No discrimination* tasks, *Ctx-dep. O1* requires high multimodal attention (animals must simultaneously attend to the visual context and auditory cues). However, no sex differences were observed in context-dependent discrimination with two outcomes (*Ctx-dep. O1/O2*), a task that shares similar attentional requirements with *Ctx-dep. O1*, but does not necessarily engage hierarchical contextual modulation. Indeed, in *Ctx-dep. O1/O2*, both the visual context and the auditory target cue are relevant, however the use of differential outcomes presumably allows animals to solve the task via simple (nonhierarchical) summation processes, rats simply adding up the converging context- and cue-evoked predictions. The possibility

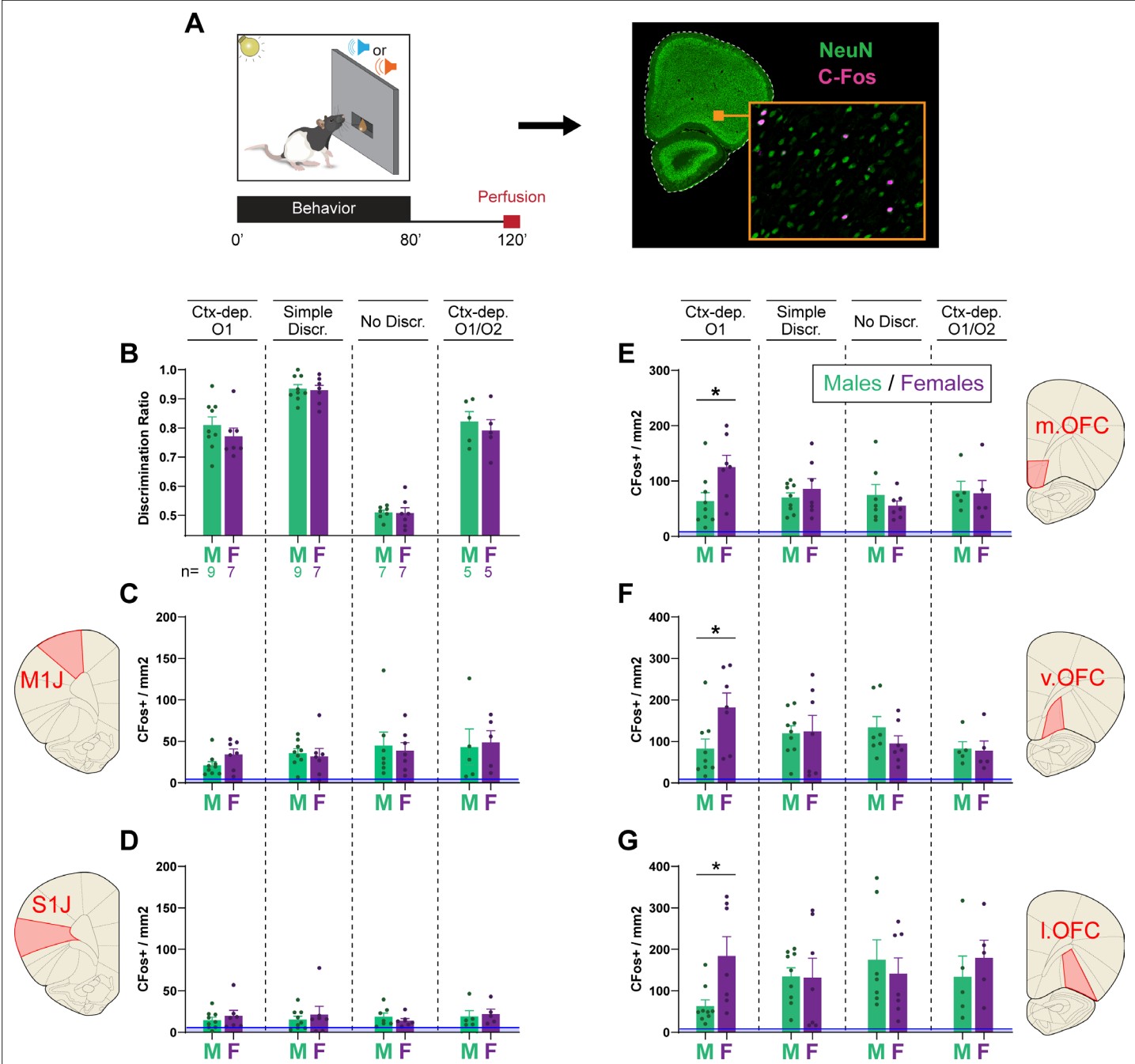

**Figure 5.** Female rats show higher orbitofrontal engagement during context-gated reward seeking. (**A**) Experimental design. Brains were collected 2 hr after the start of the behavioral session. Brain sections were stained for c-Fos (activity marker) and NeuN (general neuronal marker). (**B**) Discrimination ratios during the final behavioral session, before tissue collection for male and female rats trained in each task. c-Fos+ density in the primary jaw motor cortex (**C**), primary jaw somatosensory cortex (**D**), medial orbitofrontal cortex (**E**), ventral orbitofrontal cortex (**F**), and lateral orbitofrontal cortex (**G**) for males and females trained in each discrimination task. Error bars indicate ± standard error of the mean (s.e.m.). Blue lines (close to floor level) indicate average c-fos+ densities for animals who remained in their homecage prior to tissue collection (homecage controls; $n = 5$). *$p < 0.05$ males vs. females, $t$-tests.

The online version of this article includes the following source data for figure 5:

**Source data 1.**

of a nonhierarchical solution to the discrimination problem might explain why almost all rats trained in the *Ctx-dep. O1/O2 task* learned the discrimination and at much faster rate compared to *Ctx-dep. O1* (but see *Delamater et al., 2010* for alternative explanation of the differential outcome effect). Together, these results indicate that the sex differences observed here are not attributable to simple associative, motivational, working-memory, or attentional processes, but are specific to the neuro-computational operations required for the hierarchical, contextual control of behavior. Note that there is no a priori reason to believe that the nature of these computations differ between sexes (no sex differences in the fundamental mechanisms of context-gated predictions). Rather male and female rats appear to differ slightly in the learning rate and magnitude/robustness of these computations. Moreover it remains to be determined if the sex differences in context-gated reward seeking are specific to the sensory modalities employed here or if those sex differences extend to other sensory modalities.

Sex differences in context-dependent behaviors have previously been reported in fear conditioning preparations where females often display generalized fear responses that evade precise contextual control (*Day et al., 2016*; *Greiner et al., 2019*; *Keiser et al., 2017*; *Trask et al., 2020*). Elevated baseline anxiety and fear reactivity in females might contribute to this relative failure to regulate fear responses (*Blanchard et al., 1991*; *Dalla and Shors, 2009*; but see *Scholl et al., 2019*). However, we show here that reduced contextual control in females (at least in the early stages of acquisition) is not limited to aversive behaviors but extends to appetitive, reward-seeking behaviors. This suggests a broader, domain-general, delay in the engagement of contextual, top–down regulation processes in female rodents. This is consistent with higher sign-tracking tendencies in female rats (*Stringfield et al., 2019*; *Peterson et al., 2020*; *Fuentes et al., 2018*; *Pitchers et al., 2015*; *Hughson et al., 2019*) – another form of appetitive behavior that reflects reduced top–down cognitive control (opposed to goal-tracking behavior) (*Sarter and Phillips, 2018*; *Flagel and Robinson, 2017*; *María-Ríos and Morrow, 2020*). Note that the reduced top–down contextual control displayed by females in our study was limited to delayed acquisition. Provided sufficient training, females actually established more robust contextual control over cue-evoked reward seeking than males.

The biological basis for the delayed acquisition of contextual control in females is unknown. Although estrous cycle had no effect on the *expression* of context-dependent behavior in female discriminators, fluctuating sex hormones (and resulting neurobiological changes) could have contributed to the slower *acquisition* of contextual control in females. Indeed, learning the hierarchical contextual rules for reward prediction is a complex and lengthy process that probably requires distributed plasticity across several brain regions including the striatum, amygdala, hippocampus, and neocortex (*Fraser and Holland, 2019*; *Lee and Lee, 2013*). Estrous cycle affects synaptic physiology and plasticity at all these sites via changes in dopamine reuptake (*Morissette and Di Paolo, 1993*; *Calipari et al., 2017*), neuronal excitability (*Alonso-Caraballo and Ferrario, 2019*; *Clemens et al., 2019*; *Shanley et al., 2023*; *Scharfman et al., 2003*; *Blume et al., 2017*), spine density (*Woolley et al., 1990*; *Beeson and Meitzen, 2023*), gene expression (*Knoedler et al., 2022*; *Duclot and Kabbaj, 2015*), or neurogenesis (*Mahmoud et al., 2016*). This dynamic neurobiological environment might constitute a challenge for females attempting to solve the *Ctx-dep. O1* task, and might contribute to their delayed acquisition of contextual control. Interestingly, this dynamic neurobiological environment might also contribute to the more stable performance ultimately achieved by females. Indeed, in deep learning models, the injection of random 'noise' in (artificial) neural networks during training is commonly used to build redundancy and increase network performance and stability (*Srivastava et al., 2014*; *Zhang et al., 2021*). This '*robustness through perturbations*' approach often comes at the expense of learning speed, similar to what we observed in females. Alternatively, or in addition, androgens (male sex hormones) – known to contribute to the male advantage on certain spatial tasks – might have contributed to males' faster acquisition of contextual control in this task (*Dohanich et al., 2009*; *Spritzer et al., 2011*). The role of these hormonal factors, as well as potential non-gonadal sex differences, in the trade-off between learning speed and performance stability remains to be confirmed experimentally.

Regardless of the precise mechanism, our results indicate that, compared to male rats, female discriminators ultimately achieved more stable contextual control over cued reward seeking; their context-dependent discrimination performance was less affected by trial history (recent rewards) or by acute stress. This relative resilience of females trained in the *Ctx-dep. O1* task is consistent with

studies showing that females, while more sensitive to the arousing and anxiogenic effects of stress, are more resilient to the disrupting effects of stress on executive functions and memory (*Bangasser et al., 2018*; *Papaleo et al., 2012*). Sex differences in regional sensitivity to corticotropin-releasing factor (CRF) and CRF receptor signaling might contribute to these sex differences in behavioral responses to stress (*Hupalo et al., 2019*).

The behavioral sex differences observed in this study were accompanied by neural sex differences. Compared to males, females displayed increased OFC activation during *Ctx-dep. O1*. Importantly, no behavioral or neural sex differences were observed for any other task. Together, these results strongly implicate the OFC in the sex-biased expression of contextual modulation of reward seeking. This is consistent with recent studies demonstrating a critical role of the OFC in hierarchical control over cue-evoked behaviors (*Fraser and Janak, 2023*; *Meyer and Bucci, 2016*; *Shobe et al., 2017*). More generally, the OFC is proposed to encode task structure including context-dependent contingencies (*Mizrak et al., 2021*; *Wilson et al., 2014*; *Farovik et al., 2015*). Accordingly, the OFC appears critical for decision-making when optimal choices require knowledge of the broader task structure (i.e., when stimuli cannot be taken at face value) (*Bradfield et al., 2015*; *Takahashi et al., 2011*). Although causality remains to be established, our data suggest that the increased orbitofrontal engagement displayed by females might contribute to their resilient contextual regulation of behavior.

The root causes of the behavioral and neural sex differences in context-gated reward prediction are largely unknown and not addressed in this study. In principle, all the sex differences reported here (speed of acquisition, robustness of performance, and OFC activation) might be independent and engage different mechanisms. Alternatively, some of these effects might be causally related. Our result show that, for a comparable level of context-dependent discrimination, females show an increased engagement of the OFC compared to males (higher number of c-Fos-immunoreactive neurons). This suggests a denser (less sparse) coding of context-dependent contingencies in females' OFC. Denser coding is generally associated with slower discrimination learning but superior fault tolerance (i.e., better resistance to noise or disturbance) (*Spanne and Jörntell, 2015*), exactly what we observed in females, suggesting a potential unifying mechanism for the sex differences reported here.

The contextual regulation of behavior is at the heart of cognitive control (*Chatham et al., 2012*; *Miller and Cohen, 2001*). Deficits in this process are integral to the pathophysiology of numerous neuropsychiatric disorders (*Maren et al., 2013*). Although extrapolation of animal data to humans requires caution, our findings of a superior OFC engagement by female rats during context-informed hierarchical control of reward seeking is consistent with human [f]MRI studies showing larger OFC grey matter volume in females (*Liu et al., 2020*) and superior OFC activation in females during the regulation of emotional responses (*Lee et al., 2009*; *Welborn et al., 2009*). The behavioral and neural sex differences observed here might offer insight into the biological sex differences observed in certain disorders. For instance, the initial delay in establishing contextual control observed here in females might contribute to the rapid progression of SUD or PTSD observed in women following initial drug use or trauma (*Towers et al., 2023*; *Tolin and Foa, 2006*; *Becker and Chartoff, 2019*; *Olff, 2017*). Conversely, the more stable contextual control ultimately achieved by females after extended training observed here might contribute to the better prognosis for women following behavioral and cognitive treatments for these disorders (*Olff, 2017*; *Walitzer and Dearing, 2006*; *Galovski et al., 2013*), and might be due, in part, to increased OFC engagement.

## Methods
### Subjects

Subjects were male and female Long Evans rats (Charles Rivers), 8–10 weeks at arrival. Rats were housed in same-sex pairs in a temperature-controlled vivarium (22°C) with a 12-hr light/dark cycle; behavioral experiments were conducted during the light cycle. Rats were given free access to water in their homecage for the duration of the experiments. After a week of acclimation to the colony room, rats were mildly food restricted to maintain ~95% of age-matched free-feeding weights. All experimental procedures were conducted in accordance with UCSB Institutional Animal Care and Use Committees and the US National Institute of Health guidelines.

## Apparatus

Behavioral training was conducted in 12 identical conditioning chambers enclosed in individual sound- and light-attenuating cubicles (Med Associates, St. Albans, VT). Six chambers were dedicated to male subjects and six chambers were dedicated to female subjects; animals of both sexes were trained simultaneously. A recessed reward delivery port was located at the center of the front panel and was connected to a pellet dispenser and a syringe pump for liquid reward delivery (syringe pump located outside of the sound-attenuating cubicle). Subjects' presence in the reward port was detected by the interruption of an infrared beam. Two ceiling-facing chamber lights located on the front and back internal walls of the cubicle provided diffuse chamber illumination and were used to manipulate the background visual context. Auditory stimuli were delivered via a white noise speaker and a clicker (76 dB each), located in the back and front panel, respectively. A fan mounted on the cubicle provided ventilation and low background noise. A computer equipped with Med-PC software (Med Associates) controlled all experimental programs and data collection.

## Rewards

Food rewards consisted of a 45-mg banana-flavored food pellet (BioServe Dustless Precision Pellets, F0059 formula) or a 0.17-ml bolus of sweetened chocolate milk (2 parts chocolate Nesquik + 1 part water + 5% sucrose wt/vol) delivered over 3 s. These rewards are highly discriminable but equicaloric and were equally preferred by rats in preliminary studies. The identity of the rewarding outcomes was counterbalanced within behavioral tasks and within sexes.

## Discrimination training

Rats were randomly assigned to one of four Pavlovian discrimination conditions: Context-dependent discrimination with single outcome (*Ctx-dep. O1*, n = 30F + 32M), Context-dependent discrimination with dual outcome (*Ctx-dep. O1/O2*, n = 14F + 14M), Simple discrimination (*Simple*, n = 12F + 22M), or No Discrimination (*NoDiscr.*, n = 8F + 8M). Animals were individually placed in the conditioning chambers for 80 min training sessions 5–6 days/week. During each session, the visual background – the local context – alternated every 2 min between flashing houselights (0.25 s on, 0.25 s off; LIGHT context) or darkness (DARK context). Within each context phase, two 10 s auditory cues, X or Y (white noise or clicker, counterbalanced), were presented individually, in two separate trials, and were potentially followed by reward. The presentation of each cue within a context phase was selected semi-randomly, so that the identity of the first cue had no incidence on the identity of the second cue (i.e., the same cue could repeat twice within a context phase, so the outcome of the first trial did not inform the outcome of the second trial). Each session consisted of 40 context cycles and 80 cue presentations (i.e., 80 trials) with an average intertrial of 50 ± 20 s (rectangular distribution). The reward contingencies in each task are outlined in *Figure 1B*. During the acquisition period, unless specified otherwise (cf. *Ctx-dep. O1*), each behavioral training session consisted of an equal proportion of each trial type (20* LIGHT: X; 20* DARK: X; 20* LIGHT: Y; 20* DARK: Y).

### Context-dependent discrimination with single outcome (Ctx-dep. O1)

In this task, the reward-predictive validity of each cue was informed by the context, so that cue X was rewarded only in LIGHT context and cue Y was rewarded only in the DARK context. All rewarded trials resulted in the same reward outcome (O1). Considered individually, both cues and both contexts have equal probabilities of resulting in the same outcome, therefore accurate performance in this task cannot be understood in terms of summed associative strength between cues and contexts. Instead, accurate performance in this task critically relies on contextual modulation (i.e., occasion-setting). During the first 35 training sessions, rats experienced an equal number of rewarded and unrewarded trials (20 trials for each context/cue combination per session). On the 36th session and for the remainder of the experiment, the relative proportion of rewarded trials was reduced (10* LIGHT: X+; 30* DARK: X−; 30* LIGHT: Y−; 10* DARK: Y+) in an effort to promote discrimination. After 72 training sessions, animals that failed to reach a discrimination criterion were excluded from the rest of the study (discrimination criterion: time in port during rewarded cue ≥1.5* time in port during non-rewarded cue, for both cues, for a minimum of four out of five consecutive sessions).

### Simple discrimination

Cue X was always rewarded and cue Y was never rewarded, regardless of the context. Presumably, this training protocol promotes accurate reward prediction via simple associative learning (context-independent predictions).

### No discrimination

Both cues X and Y were probabilistically rewarded in both context (the background context provides no disambiguating information). During the acquisition period, the probability of reinforcement was 50% for each cue/context combination. Presumably, this training protocol promotes reward prediction via simple associative learning (context-independent predictions) although the reward cues retain some ambiguity.

### Context-dependent discrimination with dual outcome (Ctx-dep. O1/O2)

This task followed similar contingencies as *Ctx-dep. O1*, with the exception that the two rewarded trial types (LIGHT: X and DARK: Y) resulted in two different food outcomes (O1 and O2, respectively). This has profound implications for the learning processes engaged in the task. The delivery of different outcomes in different contexts allows for distinct context–outcome associations (LIGHT → O1 and DARK → O2), as well as distinct cue–outcome associations (X → O1 and Y → O2). These different associative structures could potentially facilitate discrimination performance via simple summation process.

## Assessment of estrous cycle

Estrous cycle phases were monitored via vaginal cytology for a subset of females ($n$ = 10) over 15 consecutive days (sessions 66–80). Immediately after the behavioral session, the tip of a saline-moistened cotton swab was inserted into the vagina and rotated to dislodge cells from the vaginal wall. Swabs were immediately rolled onto a glass slide, and the samples were preserved with a spray fixative (M-Fix, MilliporeSigma) without allowing the cells to dry. Slides were then stained with a modified Papanicolau staining procedure as follows: 50% ethyl alcohol, 3 min; tap water, 10 dips (×2); Gill's hematoxylin 1, 6 min; tap water, 10 dips (×2); Scott's water, 4 min; tap water, 10 dips (×2); 95% ethyl alcohol, 10 dips (×2); modified orange-greenish 6 (OG-6), 1 min; 95% ethyl alcohol, 10 dips; 95% ethyl alcohol, 8 dips; 95% ethyl alcohol, 6 dips; modified eosin azure 36 (EA-36), 20 min; 95% ethyl alcohol, 40 dips; 95% ethyl alcohol, 30 dips; 95% ethyl alcohol, 20 dips; 100% ethyl alcohol, 10 dips (×2); xylene, 10 dips (×2); coverslip immediately. Samples were observed with light microscopy (×4). Four criteria were used to determine the specific estrus cycle: (1) cell density, (2) percentage of nucleated epithelial cells, (3) cornified epithelial cells, and (4) percentage of leukocytes (*Cora et al., 2015*; *Goldman et al., 2007*; *Karim et al., 2003*). Each sample was independently rated by two trained observers (S.P. and A.M.), blind to the behavioral data. Inter-rater agreement was high (88.2%). Only behavioral sessions for which both raters agreed on the estrous phase were selected for analysis. As a result, two to four behavioral sessions were considered for each subject at each phase of the estrous cycle. Behavioral data on these sessions were averaged to express a single data point for each rat at each stage of the estrous cycle. Females that did not show a regular cycle were excluded from the analysis ($n$ = 2).

## Acute restraint stress

Before testing the effect of acute stress on discrimination performance, the relative proportion of rewarded trials was adjusted in all groups to match the proportion of rewarded trials in the context-dependent discrimination with a single outcome (*Ctx-dep. O1*). As a result, rewarded cues became less frequent and represented only 1/4 of the trials. For all groups, these rewarded trials were equally distributed across both contexts. After five to six sessions of adjustment to these new conditions, we tested the effect of an acute stress on discrimination performance in a subset of animals from all training groups (Ctx-dep. O1: M $n$ = 15, F $n$ = 11; Simple discrimination: M $n$ = 6, F $n$ = 6; No discrimination: M $n$ = 8, F $n$ = 8; Ctx-dep. O1/O2: M $n$ = 7, F $n$ = 7). Acute stress was induced by restraining rats in standard Plexiglas rat restraint tubes (Plas Labs, Inc) for 90 min, in a novel brightly lit room. To account for the size differences between male and females, two different sized restrainers were used (65 and 50 mm internal diameter for male and female restrainers, respectively). After placing the rats

in the tube, the length of the restrainer was adjusted to immobilize the rat without causing pain or interfering with animals' breathing. After 90 min of restraint, the rats were returned to their homecage and left undisturbed for 5 min before being placed in the conditioning chambers for a normal behavioral session. This restraint procedure is a well-established stressor in rodents, known to produce robust behavioral and physiological effects (*Shansky et al., 2006*; *Babb et al., 2013*; *Lovelock and Deak, 2020*).

## cFos immunofluorescence

Following stress test, animal were retrained for five to six sessions, under standard conditions (no stress; equal number of rewards for all training groups; rewarded trials = 1/4 of all trials). After their last behavioral session, animals remained in the conditioning chambers for 40 min (120 min total); they were then deeply anesthetized with a lethal dose of Euthasol and perfused transcardially with ice-cold 1× phosphate-buffered saline (PBS) followed by ice-cold 4% paraformaldehyde (PFA) in phosphate buffer (PB). Brains were extracted, post-fixed (4% PFA overnight) and cryoprotected (30% sucrose + 0.1% sodium azide, 1× PBS) until processed for immunostaining. Brains were frozen and sectioned into 40 µm with a sliding cryostat (Leica CM1860). Free floating coronal sections were washed for 10 min in PBS, followed by a 2-hr incubation in a blocking solution (5% Normal Goat Serum [NGS], 0.2% Triton X-100 in 1× PBS) at room temperature. Sections were then incubated with rabbit anti-cFos (1:1000, Cell Signaling, CAT#: 2250) and mouse anti-NeuN (neuron-specific protein; 1:2000 or 1:3000, Novus, CAT#: NBP1-92693) primary antibodies, in the blocking solution, for 20 hr at 4°C. Following primary antibody incubation, sections were washed in PBS three times for 10 min and then incubated with anti-rabbit Alexa Fluor 647-conjugated secondary antibody (far-red) (1:250, Jackson ImmunoResearch, CAT#: 111-605-144) and anti-mouse Alexa Fluor 488-conjugated secondary antibody (green) (1:500, Jackson ImmunoResearch, CAT#: 115-545-003) for 90 min at room temperature. Sections were then washed in PBS three times for 10 min, mounted on slides and immediately coverslipped with a hardset mounting medium (Vector laboratories CAT#: H-1700). Images were acquired at a ×10 magnification, using a Keyence BZ-X fluorescence microscope.

Brain regions of interest for c-Fos-immunoreactive neurons quantification included the medial, ventral, and lateral orbitofrontal cortex (mOFC, vOFC, and lOFC). Additionally, two control regions were included, for which we did not expect task-specific activation: the jaw primary motor (M1J) and somatosensory cortex (S1J). These regions of interests were manually isolated from the original images, using only the green/NeuN channel (the far-red/c-fos channel was temporarily disabled for blinding purposes). A custom ImageJ script (National Institutes of Health) was then used to count c-Fos-positive neurons in the selected regions and normalize this count to the surface area of the selected region (c-Fos-positive neurons/mm$^2$). For each brain region, c-Fos-positive neurons were counted bilaterally from 2 to 4 sections per animal and a mean count was computed for each animal.

## Statistical analysis

Data were collected over three replication cohorts, each composed of approximately equal number of male and female rats. Cue-evoked reward seeking was quantified primarily as the percentage of time a rat spent in the food cup during the last 5 s of cue presentation, when anticipatory food cup behavior is more reliably observed and less contaminated by orienting behaviors (*Holland, 1977*; *Holland, 1980*; *Holland, 2000*). The percentage of time in port during the 5 s epoch prior cue onset is shown but not analyzed. Discrimination accuracy was expressed as a discrimination ratio, defined as the time in port during rewarded trials divided by the total time in port during all trials. A ratio of 0.5 indicates equal responding during rewarded and non-rewarded trials (i.e., no discrimination) whereas a ratio of 1 indicates responding exclusively during rewarded trials (i.e., perfect discrimination). Time in port and discrimination ratio during acquisition were aggregated in four-session bins. These longitudinal RM were analyzed with LMM with a Restricted Maximum Likelihood estimation method. All covariance structures were explored and the best fitting model was determined by the lower Akaike information criterion score (*Duricki et al., 2016*; *West, 2009*). The selection of a covariance structure influenced the degrees of freedom of the model, possibly resulting in non-integer values. The remaining of the analyses consisted generally of mixed-models RM-ANOVAs with Sex and/or Task as a between-subject factor, and Trial history, Stress, or Brain region as within-subject factors. The Geisser–Greenhouse correction was applied when data lacked sphericity, possibly resulting in

non-integer degrees of freedom. Post hoc or planned contrast pairwise comparisons were carried with Bonferroni-corrected *t*-tests. When appropriate, nonparametric tests were conducted and consisted of the Kruskal–Wallis *H* test followed by post hoc Bonferroni-corrected Dunn's pairwise comparisons. All tests were two-tailed, significance was assessed against a type I error rate of 0.05. Statistical analyses were conducted with SPSS Statistics package (version 28.0.0.0; IBM SPSS). Graphs were generated using GraphPad Prism 9.

## Acknowledgements

This work was supported by UC Santa Barbara Academic Senate Research Grant.

## Additional information

### Funding

| Funder | Grant reference number | Author |
|---|---|---|
| University of California, Santa Barbara | Academic Senate | Ronald Keiflin |

The funders had no role in study design, data collection, and interpretation, or the decision to submit the work for publication.

### Author contributions

Sophie Peterson, Amanda Maheras, Conceptualization, Formal analysis, Investigation, Methodology, Writing – original draft, Writing – review and editing; Brenda Wu, Jose Chavira, Investigation, Methodology; Ronald Keiflin, Conceptualization, Supervision, Funding acquisition, Investigation, Visualization, Methodology, Writing – original draft, Project administration, Writing – review and editing

### Author ORCIDs

Brenda Wu ⓘ https://orcid.org/0000-0003-4921-8628
Ronald Keiflin ⓘ https://orcid.org/0000-0001-5347-7337

### Ethics

All experimental procedures were conducted in strict accordance with the Guide for the Care and Use of Laboratory Animals of the National Institutes of Health. The protocol was approved by the Institutional Animal Care and Use Committee (IACUC) of the University of California, Santa Barbara (protocol# 951).

Reviewer #1 (Public review): https://doi.org/10.7554/eLife.93509.4.sa1
Reviewer #2 (Public review): https://doi.org/10.7554/eLife.93509.4.sa2
Reviewer #3 (Public review): https://doi.org/10.7554/eLife.93509.4.sa3
Author response https://doi.org/10.7554/eLife.93509.4.sa4

## Additional files

### Supplementary files
- MDAR checklist
- Source code 1. ImageJ macro for cFos count per surface area.

### Data availability

All data generated or analyzed during this study are included in the manuscript and supporting files; source data files have been provided for Figures 1–5 and Figure 2—figure supplement 1.

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
