## [Editor Report · eLife assessment]

This **valuable** manuscript reveals sex differences in bi-conditional Pavlovian learning and conditional behavior. Males learn hierarchical context-cue-outcome associations more quickly, but females show more stable and robust task performance. These sex differences are related to cellular activation in the orbitofrontal cortex. Although the evidence supporting these claims is **convincing**, some assertions of sex differences in context-dependent discrimination behaviour may be slightly overstated yet have strong potential to guide future research to clarify the nature of these differences. The results will be of interest to many behavioural neuroscientists, particularly those who investigate sex-specific behaviours.

---

## [Referee Report · Reviewer #1 (Public review)]

Summary:

Peterson et al., present a series of experiments in which the Pavlovian performance (i.e. time spent at a food cup/port) of male and female rats is assessed in various tasks in which context/cue/outcome relationships are altered. The authors find no sex differences in context-irrelevant tasks, and no such differences in tasks in which the context signals that different cues will earn different outcomes. They do find sex differences, however, when a single outcome is given and context cues must be used to ascertain which cue will be rewarded with that outcome (Ctx-dep O1 task). Specifically, they find that males acquired the task faster, but that once acquired, performance of the task was more resilient in female rats against exposures to a stressor. Finally, they show that these sex differences are reflected in differential rates of c-fos expression in all three subregions of rat OFC, medial, lateral and ventral, in the sense that it is higher in females than males, and only in the animals subject to the Ctx-dep O1 task in which sex differences were observed.

Strengths:

• Well written

• Experiments elegantly designed

• Robust statistics

• Behaviour is the main feature of this manuscript, rather than any flashy techniques or fashionable lab methodologies, and luckily the behaviour is done really well.

• For the most part I think the conclusions were well supported, although I do have some slightly different interpretations to the authors in places.

Weaknesses:

The authors have done an excellent job of addressing all previous weaknesses. I have no further comments.

---

## [Referee Report · Reviewer #2 (Public review)]

Summary:

A bidirectional occasion-setting design is used to examine sex differences in the contextual modulation of reward-related behaviour. It is shown that females are slower to acquire contextual control over cue-evoked reward seeking. However, once established, the contextual control over behaviour was more robust in female rats (i.e., less within-session variability and greater resistance to stress) and this was also associated with increased OFC activation.

Strengths:

The authors use sophisticated behavioural paradigms to study the hierarchical contextual modulation of behaviour. The behavioural controls are particularly impressive and do, to some extent, support the specificity of the conclusions. The analyses of the behavioural data are also elegant, thoughtful, and rigorous.

Weaknesses:

The authors have addressed the major weaknesses that I identified in a previous review.

---

## [Referee Report · Reviewer #3 (Public review)]

Summary:

This manuscript reports an experiment that compared groups of rats acquisition and performance of a Pavlovian bi-conditional discrimination, in which the presence of one cue, A, signals that the presentation of one CS, X, will be followed by a reinforcer and a second CS, Y, will be nonreinforced. Periods of cue A alternated with periods of cue B, which signaled the opposite relationship, cue X is nonreinforced and cue Y is reinforced. This is a conditional discrimination problem in which the rats learned to approach the food cup in the presence of each CS conditional on the presence of the third background cue. The comparison groups consisted of the same conditional discrimination with the exception that each CS was paired with a different reinforcer. This makes the problem easier to solve as the background is now priming a differential outcome. A third group received simple discrimination training of X reinforced and Y nonreinforced in cues A and B, and the final group were trained with X and Y reinforced on half the trials (no discrimination). The results were clear that the latter two discrimination learning procedures resulted in rapid learning in comparison to the first. Rats required about 3 times as many 4-session blocks to acquire the bi-conditional discrimination than the other two discrimination groups. Within the biconditional discrimination group, female and male rats spent the same amount of time in the food cup during the rewarded CS, but females spent more time in the food cup during CS- than males. The authors interpret this as a deficit in discrimination performance in females on this task and use a measure that exaggerates the difference in CS+ and CS_ responding (a discrimination ratio) to support their point. When tested after acute restraint stress, the male rats spent less time in the food cup during the reinforced CS in comparison to the female rats, but did not lose discrimination performance entirely. The was also some evidence of more fos positive cells in the orbitofrontal cortex in females. Overall, I think the authors were successful in documenting performance on the biconditional discrimination task, showing that it is more difficult to perform than other discriminations is valuable and consistent with the proposal that accurate performance requires encoding of conditional information (which the authors refer to as "context"). There is evidence that female rats spend more time in the food cup during CS-, but this I hesitate to agree that this is an important sex difference. There is no cost to spending more time in the food cup during CS- and they spend much less time there than during CS+. Males and females also did not differ in their CS+ responding, suggesting similar levels of learning, A number of factors could contribute to more food cup time in CS-, such as smaller body size and more locomotor activity. The number of food cup entries during CS+ and CS- was not reported here. Nevertheless, I think the manuscript will make a useful contribution to the field and hopefully lead readers to follow up on these types of tasks. One area for development would be to test the associative properties of the cues controlling the conditional discrimination, can they be shown to have the properties of Pavlovian occasion setting stimuli? Such work would strengthen the justification/rationale for using the term "context" and "occasion setter" to refer to these stimuli in this task in the way the authors do in this paper.

Strengths:

Nicely designed and conducted experiment.

Documents performance difference by sex.

Weaknesses:

Overstatement of sex differences.

Inconsistent, confusing, and possibly misleading use of terms to describe/imply the underlying processes contributing to performance.

---

## [Author Response]

The following is the authors’ response to the previous reviews.

**Reviewer 2 (Public Review):**
Stress response in males versus females: The authors argue that the contextual control over behaviour was more robust in female rats as females show less within session variability and greater resistance to stress. What evidence is there that the restraint stress procedure caused a similar stress response in both sexes? That is, was the stress induction equally effective in males and females?

The restraint protocol used in this study is a well-established stressor in rodents, known to produce robust behavioral and physiological effects (HPA axis activation), in both sexes. Although not measured in this study, the ACTH and cortisol responses are actually greater in females during restraint. To the extent that “stress induction” is interpreted as “HPA axis activation”, this strongly suggests that the stress induction in males and females was at least comparable, if not greater in females.

We have added a few sentences (in the Result and Method section) to highlight this important point. We thank the reviewer for bringing this up.

Minor corrections:(1) Please verify that the in-text reference to the figures is correct. I noticed a few mistakes, for example:- Line 120 (pdf) refers to Fig. 1 C-D but should refer to D only.- Line 312 (pdf) refers to Fig 1D for discrimination ratios but these are shown in Fig 1E- No reference in text to 2A

Thank you for bringing this to our attention. We have fixed the in-text references to the figures.

(2) In the results it states that the homecage c-Fos+ counts are shown in Figure 5 but I couldn't see these?

The homecage c-Fos+ counts were initially shown as a pale gray band in the background of the main histograms. Because those counts are very low, it was hard to dissociate this gray band from the black horizontal axis. We have replaced the gray band with a more vivid blue line that is now in the foreground of the histograms. Moreover, we added a note in the figure legend to bring readers’ attention to this homecage count line, close to floor level.

(3) Line 306: It is stated that "the use of differential outcomes presumably allows animals to solve the task via simple (nonhierarchical) summation processes". I don't understand the use of "summation" here, isn't it simply that the rats are relying on direct context-outcome and/or cue-outcome associations?

That’s right. These rats might be relying on direct context-outcome and cue-outcome associations and adding (or summing up) the converging expectations. We have added a few words in the text to clarify what we mean by summation (i.e. the addition of converging cue-evoked + context-evoked predictions).